# Effect of Live and Fragmented *Saccharomyces cerevisiae* in the Feed of Pigs Challenged with *Mycoplasma hyopneumoniae*

**DOI:** 10.3390/pathogens13040322

**Published:** 2024-04-14

**Authors:** Gabriela Vega-Munguía, Alejandro Vargas Sánchez, Juan E. Camacho-Medina, Luis Suárez-Vélez, Gabriela Bárcenas-Morales, David Quintar Guerrero, Abel Ciprian-Carrasco, Susana Mendoza Elvira

**Affiliations:** Department of Biological Sciences, Facultad de Estudios Superiores Cuatitlan, Universidad Nacional Autónoma de México, Cuautitlán Izcalli 54720, Mexico; anagabrielvm@hotmail.com (G.V.-M.); jecm_iztla@yahoo.com.mx (J.E.C.-M.); luedsuve@gmail.com (L.S.-V.); gbarcenas2019@gmail.com (G.B.-M.); quintana@unam.mx (D.Q.G.); abelciprian47@gmail.com (A.C.-C.)

**Keywords:** *Mycoplasma hyopneumoniae*, pigs, live *S. cerevisiae*

## Abstract

Currently, the responsible use of antimicrobials in pigs has allowed the continuous development of alternatives to these antimicrobials. In this study, we describe the impact of treatments with two probiotics, one based on live *Saccharomyces cerevisiae* (*S. cerevisiae*) and another based on fragmented *S. cerevisiae* (beta-glucans), that were administered to piglets at birth and at prechallenge with *Mycoplasma hyopneumoniae*. Thirty-two pigs were divided into four groups of eight animals each. The animals had free access to water and food. The groups were as follows: Group A, untreated negative control; Group B, inoculated by nebulization with *M. hyopneumoniae* positive control; Group C, first treated with disintegrated *S. cerevisiae* (disintegrated *Sc*) and inoculated by nebulization with *M. hyopneumoniae*; and Group D, treated with live *S. cerevisiae* yeast (live *Sc*) and inoculated by nebulization with *M. hyopneumoniae*. In a previous study, we found that on Days 1 and 21 of blood sampling, nine proinflammatory cytokines were secreted, and an increase in their secretion occurred for only five of them: TNF-α, INF-α, INF-γ, IL-10, and IL-12 p40. The results of the clinical evolution, the degree of pneumonic lesions, and the productive parameters of treated Groups C and D suggest that *S. cerevisiae* has an immunomodulatory effect in chronic proliferative *M. hyopneumoniae* pneumonia characterized by delayed hypersensitivity, which depends on the alteration or modulation of the respiratory immune response. The data presented in this study showed that *S. cerevisiae* contributed to the innate resistance of infected pigs.

## 1. Introduction

*Mycoplasma hyopneumoniae* (*M. hyopneumoniae*) is the principal etiological agent responsible for enzootic pneumonia (EP) in pigs, and other secondary pathogens are also frequently involved in the porcine respiratory complex (PRC) [1]. In the treatment and control of enzootic pneumonia, strategic medication with active antimicrobials is considered, preferably against *M. hyopneumoniae*, but it can also be useful against the main secondary bacteria, especially during periods in which pigs are at risk of respiratory diseases [2]. There are a number of studies on the efficacy of antibiotics evaluated in vitro, and the results show good susceptibility against *M. hyopneumoniae* [1]. Although the clinical effects of antibiotics against *M. hyopneumoniae* have also been evaluated in vivo, they are administered through the ingestion of premixes in food [3,4,5].

Growth-promoting antibiotics have been used in piglet diets to reduce the risk of digestive tract diseases, mortality, and morbidity, and the responsible use of antimicrobials in pigs and the continuous development of antimicrobial alternatives are priorities for ensuring sustainable, long-term development of pigs [6]. The administration of antibiotics to animals has been the subject of intense scrutiny due to its contribution to the spread of antibiotic-resistant bacteria in the food chain [7]. In the past, the use of antibiotics as additives (growth-promoting antibiotics—GPAs) in the feed of poultry and pigs was a way to increase animal growth rates and limit the development of infections by pathogenic microorganisms. However, this use was conditioned on searching for new alternative components for food, which would ensure good health and animal production.

The use of probiotics as feed additives for pigs has proven to be a solution that is addressing expectations in animal production and is accepted by consumers [7,8,9]. The prohibition of GPAs in the diet of pigs has required the development of alternative strategies, such as the administration of different strains of *Saccharomyces cerevisiae* to increase the concentrations of immunoglobulins in both pig colostrum and milk and decrease the incidence of diarrhea during nursing [10,11]. Additionally, in other studies conducted in the last stages of pregnancy and lactation, it was found that the administration of *S. cerevisiae* to sows increased the content of gamma globulin in their milk, while the administration of yeast mannan oligosaccharides increased the concentration of IgG in their colostrum [12,13,14]. In another strategy used by Kiros et al. [15], we compared the performance and the microbiota of the hindgut of weaned piglets, which were subjected to different regimens of live yeast supplements of *S. cerevisiae* (Actisaf Sc 47) before and after weaning. The effect of yeast supplementation resulted in the development of phylogenetically homogeneous and less dispersed microbial communities compared to the microbiota of control piglets. However, the authors indicated that the potential mechanisms by which yeast supplementation can modulate piglet performance are not yet well understood.

It has been found that probiotics also have an immunomodulatory effect, although their mechanism of action is not yet fully understood. Probiotics also stimulate immunocompetent cells to produce cytokines that improve, reduce, or regulate the local immune system and its response [16,17]. Studies have mainly been conducted in humans and have shown that the immunomodulatory effect of the studied probiotic triggered an increase in the activity of phagocytic cells and that the variability depended on the species and strains of bacteria, the viability of the bacteria, and the dose of the probiotic preparation used [18,19,20].

It is thought that the potential immunomodulatory effect of probiotics is involved in maintaining the balance of Th1/Th2 lymphocytes in cooperation with Th17 lymphocytes and regulatory T lymphocytes (T_reg_) [21,22,23]. In this study, we describe the impact of two probiotic treatments, one based on live *S. cerevisiae* and another based on fragmented *Saccharomyces cerevisiae* (beta-glucans), that were administered to piglets at birth and at prechallenge with *M. hyopneumoniae*.

## 2. Materials and Methods

Animals. The animals were obtained from the maternity units of a *M. hyopneumoniae*-negative farm at the Center for Teaching, Research, and Extension in Porcine Production (CEIEPP) of the National Autonomous University of Mexico (UNAM), located in Jilotepec, state of Mexico. Newborn piglets were selected, obtained from four different litters, from fifth-parity sows of the Yorkshire breed. The piglets were free of antibodies to *M. hyopneumoniae*, *Actinobacillus pleuropneumoniae* serotypes 1–9, and the porcine reproductive and respiratory syndrome (PRRS) virus. The piglets were fed commercial pelleted feed free of antibiotics (Vimifos SA de CV).

Experimental design. Thirty-two selected piglets were divided into four groups. The pigs in Group A negative control and Group B positive control did not receive the yeast, while the animals in Groups C (*Sc* disintegrated) and D (*Sc* live) were administered the yeast at 0.25 g/animal/day and at 0.50 g/animal/day [15], respectively, through direct intake starting on Day 0, with the yeast being subsequently included in the preinitiator feed until Day 21. Immediately after weaning, the piglets were transferred to the National Center for Diagnostic Services in Animal Health (CENASA), located in Santa Ana Tecámac, state of Mexico. Thirty-two pigs were divided into four groups of eight animals each. Each group was housed separately in four pens in the isolation unit. The animals had free access to water and food. Group A was an untreated negative control; Group B was inoculated by nebulization with the positive control *M. hyopneumoniae*; Group C was first treated with disintegrated *S. cerevisiae* (disintegrated *Sc*) and inoculated by nebulization with *M. hyopneumoniae*; and Group D was treated with live yeast *S. cerevisiae* (live *Sc*) and inoculated by nebulization with *M. hyopneumoniae*.

In the case of Groups C and D, *S. cerevisiae* continued to be administered upon arrival at the experimental units. Dilutions of the products were made in physiological saline solution, where 50 g/L of disintegrated *S. cerevisiae* (disintegrated *Sc*) were mixed per L for Group C and 100 g of live *S. cerevisiae* (*Sc* alive) per liter of physiological saline solution for Group D. The treatments were provided orally starting on Day 21 of life by forced intake for seven continuous days. For the control groups, only physiological saline solution was applied in the same amounts of liquid (5 mL per piglet) that were provided to the remaining treatments. During the challenge and experimentation, a commercial pelletized preinitiator from Trouw Nutrition and Vimifos SA de CV was used following the manufacturer’s instructions.

Microorganism. *Mycoplasma hyopneumoniae* strain 194 was donated by Dr. Richard Ross, Veterinary Medical Research Institute, College of Veterinary Medicine, Iowa State University, Ames, Iowa, in a pneumonic lung suspension.

*Culture medium*. For the culture of *M. hyopneumoniae*, Friis medium [24] was supplemented with kanamycin sulfate (1.0 mg/mL), 20% (*v*/*v*) horse serum, 10% (*v*/*v*) pork serum, and 10% (*v*/*v*) fresh yeast extract. To prepare the solid Friis medium, 0.9% (*w*/*v*) Noble agar (Difco Laboratories, Detroit, Michigan) was added.

Preparation and standardization of the inoculum. The inoculum of *M. hyopneumoniae* was cultured in 1 L flasks using liquid Friis medium at 100 rpm agitation for 7 days. To determine the titer of color-changing units (CCU), an aliquot was taken, and serial dilutions were made in fresh Friis medium up to 10^5^ and incubated between 5 and 7 days at 37 °C. The inoculum diluted to contain 10^8^ CCU/mL was used to inoculate Groups B, C, and D in an aerosol chamber using the procedure described below.

Challenge by aerosolization with the suspension of *M. hyopneumoniae*. A standardized volume of mycoplasma (10^8^ UCC/mL) was aerosolized at a constant ratio (25 mL per group/animal in 30 min) in an aerosolization chamber containing four medical nebulizers (DeVilbiss, Somerset, Pennsylvania); the camera was designed and created by Lara et al. [25]. All the animals of the 3 experimental groups were challenged with *M. hyopneumoniae* (at a UCC concentration of 1 × 10^8^ per mL) on Day 7 postweaning in the isolation units. Three nebulizations were performed with an interval of 24 h.

Clinical signs. Throughout the experiment, clinical signs of temperature, respiratory signs of cough, and dyspnea, both at rest and on exertion, were determined as previously described [26].

Necropsy. Pigs were sedated with 2 mg of azaperone (Janssen Pharmaceutica, Beerse, Belgium) per kg of body weight and deeply anesthetized with 1.5 mg of metomidate hydrochloride (Janssen Pharmaceutica, Beerse, Belgium) per kg of body weight. Pigs were sacrificed by electric shock, followed by exsanguination. The lungs were removed, and the pneumonic areas were schematized and evaluated. The extension of the pneumonic lesions was determined by planimetry in standardized diagrams of the ventral and dorsal areas of the pneumonic lungs, considering the surface of both sides as 100% [26]. Statistically significant differences between groups (*p* ≥ 0.05) were determined with “student’s *t*-test”.

Recovery of the microorganisms. Samples were collected from the pneumonic lesions and tracheal explants of the lungs from the organisms in Groups B, C, and D; the homogenates of the pneumonic and dissected areas of the tracheal explants were placed in liquid Friis medium [27,28]. *Mycoplasma hyopneumoniae* was identified by PCR [28,29].

Growth performance of pigs. The average daily weight gain (DWG) was calculated for each experimental group. Variance analysis was applied to the DWG of the groups being compared.

## 3. Results

### 3.1. Temperature

The pigs in Group A had normal temperatures during the experiment, with a mean ± SD of 39.27 ± 0.09 (see Figure 1). The pigs in Group B had two peaks of hyperthermia on Day 7 (39.8 °C) and Day 13 (39.7 °C); later, the temperature was within the normal limits. The pigs in Group C had normal temperatures during the experiment. The pigs in Group D had only one peak of hyperthermia on Day 13 (39.7 °C), but later, the temperature was within the normal limits.

### 3.2. Respiratory Signs

Pig respiration in Group A was normal, while all pigs in Group B developed a nonproductive cough from Day 11 postinoculation with *M. hyopneumoniae* that evolved from mild to moderate, with a long presentation of more than 8 days, but it decreased until Day 26 at the end of the experiment (Figure 2A). Dyspnea at rest predominated over dyspnea during exertion in all animals (Figure 2B). Of the pigs in Group C, seven developed a nonproductive cough that evolved mainly from mild to moderate and had a short presentation of less than 7 days, which decreased until the end of the experiment (Figure 3A). Exertional dyspnea predominated over resting dyspnea in all animals (Figure 3B). Of the pigs in Group D, seven developed a nonproductive cough that evolved mainly from mild to moderate and had a short presentation of less than 6 days, and it decreased until the end of the experiment (Figure 4A). Exertional dyspnea predominated over resting dyspnea in all animals (Figure 4B).

### 3.3. Pathological Pulmonary Lesions

The distribution, extension, and appearance of the macroscopic pulmonary lesions are summarized in Table 1. In untreated and unchallenged Group A, very mild pulmonary lesions were found in three lungs that ranged from 0.1 to 0.3% and were characterized by small areas of reddish consolidation on the lung surface. The pigs in Group B challenged only with *M. hyopneumoniae* had consolidated reddish gray lesions that varied in extent from 9.2 to 29% in all animals. The animals in Group C treated with disintegrated *S. cerevisiae* and challenged with *M. hyopneumoniae* presented reddish gray consolidated areas with an extension of 0.1 to 1.0% in all pigs. The pigs in Group D treated with live *S. cerevisiae* and challenged with *M. hyopneumoniae* showed reddish gray consolidation areas of variable extension of 1.4 to 4.2% in all animals, with “student’s *t*-test” test (Figure 5).

### 3.4. Distribution of Pulmonary Injuries

The distribution and appearance of lung lesions are summarized in Table 1. No pneumonic lesions were observed in Group A. In Group B, pneumonic lesions were reddish gray areas of consolidation with pleural adhesions in two cases and located in the anteroventral lobes of the lung as well as in the accessory lobes. In Group C, the lesions were distributed in the anteroventral pulmonary areas to a lesser extent than those found in Group D; in addition, the lesions had more reddish than gray consolidations, and there were no pleural adhesions (Table 2).

### 3.5. Recovery of the Microorganisms

*Mycoplasma hyopneumoniae* was not isolated from the lungs of the Group A pigs. *M. hyopneumoniae* was recovered from the homogenates of the pneumonic lesions and tracheal explants from the lungs of the pigs in Groups B, C, and D (Table 1). *Mycoplasma hyopneumoniae* was identified by PCR.

### 3.6. Growth Performance

The pigs in all groups experienced similar daily weight gain amounts in the first three stages: previous, start, and challenge. During stage 4, only the challenged groups showed a decrease in their weight; however, during the evolution of the disease, only Group B showed a decrease in weight up to the final stage of the experiment, while Groups C and D increased in weight at this stage (Figure 6).

## 4. Discussion

For many years, one way to increase the growth rate of pigs and limit the development of infections by pathogenic microorganisms was the use of antibiotics as food additives, also known as growth-promoting antibiotics (GPAs) [8,30].

The prohibition of GPAs in the diets of pigs has required the development of alternative strategies. GPAs have been used in piglet diets to reduce the risk of digestive tract diseases, and the responsible use of antimicrobials in pigs and the continuous development of antimicrobial alternatives such as colistin continue to be a priority for ensuring the sustainable development of pigs over the long term [6].

The primary effect of probiotics in the gastrointestinal tract of pigs is the prevention of mucosal colonization by pathogenic microbes. The mechanism of this probiotic effect is based on competition with pathogens for adhesion sites in enterocytes and stimulation of local and systemic immune mechanisms in the host [31,32,33].

Studies that were conducted in the last stages of pregnancy and lactation have found that the administration of *S. cerevisiae* to sows increased the content of gamma globulin in their milk, while the administration of mannan oligosaccharides from yeast increased the concentration of IgG in their colostrum [12,13,14]. The administration of different strains of *S. cerevisiae* has been used to increase the concentrations of immunoglobulins in both colostrum and milk, thus decreasing the incidence of diarrhea during lactation, but more studies are required to evaluate the antigenic specificity of immunoglobulins in both colostrum and milk [10,11].

In addition, studies that were conducted mainly in humans showed that the immunomodulatory effect of a probiotic triggered an increase in the activity of phagocytic cells and variability depending on the species and strains of bacteria, the viability of the bacterium, and the dose of the probiotic preparation used [18,19,20].

In this study, we describe the impact of treatments with two probiotics, one based on live *S. cerevisiae* and another based on fragmented *S. cerevisiae* (beta-glucans), that were administered to piglets at birth and at prechallenge with *M. hyopneumoniae*. The data presented in this study show that *S. cerevisiae* contributed to the innate resistance of infected pigs.

In the clinical study, it was observed that the pigs in Group A had normal temperatures throughout the experiment, while the pigs in Group B, which had three consecutive challenges with *M. hyopneumoniae*, had two peaks of hyperthermia on Day 7 (39.8 °C) and on Day 13 (39.7 °C); in addition, in this period, clinical respiratory signs of cough and dyspnea developed, and in piglets, a clinical form that develops between 10 and 16 days was found [34]. Subsequently, the temperatures of the pigs were maintained within the normal limits. The pigs in Group C had normal temperatures during the experiment, while the pigs in Group D had only a peak of hyperthermia on Day 13 (39.7 °C), and subsequently, their temperatures remained within the normal limits. These animals developed clinical signs of moderate respiratory symptoms. The respiratory signs in Group A pigs were normal. However, all pigs in Group B developed a nonproductive cough after inoculation with *M. hyopneumoniae*, which evolved from mild to moderate with a long presentation that decreased until the end of the experiment, and dyspnea occurred when all animals with chronic behaviors exerted effort [34]. The pigs treated with disintegrated *Sc* in Group C and those treated with live *Sc* in Group D and challenged with *M. hyopneumoniae* presented nonproductive coughs that evolved from mild to moderate with short durations, which decreased until the end of the experiment, and the remaining animals exhibited dyspnea at rest. The evolution of the respiratory signs of cough and dyspnea in Groups C and D was of a lower degree, with a tendency to recover and normal appearances [1]. These results allowed us to determine that the pigs in Groups C and D that were treated with both disintegrated and live *S. cerevisiae* developed a protective effect against the challenge of *M. hyopneumoniae* due to the minimal clinical respiratory signs that they presented.

In relation to the pathological pulmonary lesions found in the groups challenged with *M. hyopneumoniae*, the pigs in the untreated and unchallenged Group A had very mild pulmonary lesions in three lungs; the extension varied between 0.1% and 0.3%, and they appeared as small areas of reddish consolidation on the lung surface. The pigs in Group B that were only challenged with *M. hyopneumoniae* had consolidated reddish gray lesions in all animals that varied in extension from 9.2% to 29%, located mainly in the cranioventral areas of the pulmonary lobes, characteristic of chronic proliferative pneumonia. The animals in Groups C and D treated with disintegrated and live *S. cerevisiae*, respectively, and challenged with *M. hyopneumoniae* presented reddish gray consolidated areas with extensions of 0.1% to 1.0% in some Group C pigs and 1.4 to 4.2% in Group D pigs. (see Table 1). In this study, the treatment protocols for the piglets in Groups C and D followed an approach that was applied in a previous study [35], and cytokines were not determined.

In a previous study carried out by our group, the yeast *S. cerevisiae* was administered in its two presentations, *Sc* disintegrated and *Sc* live, to newborn piglets on Day 1 until weaning on Day 21. Blood samples were taken only on Days 1 and 21, and the secretion of nine cytokines was evaluated: TNF-α, INF-α, INF-γ, IL-1β, IL-4, IL-6, IL-8, IL-10, and IL-12-p40. An increase in the secretion of five proinflammatory cytokines, TNF-α, INF-α, INF-γ, IL-10, and IL-12 p40, was found [35]. Additionally, when evaluating the local and systemic immune response induced by a bacterin of *M. hyopneumoniae* in pigs “in vivo”, it was found that macrophages produce proinflammatory cytokines that include IL-1, IL-6, IL-8, and tumor necrosis factor alpha (TNFα), while in a dual infection with *M. hyopneumoniae* and the PRRS virus in tracheobronchial washes of pigs experimentally infected with mycoplasma until Day 28, an increase in IL-10 and IL-12 was found [36,37].

In a dynamic study of an experimental infection with *M. hyopneumoniae* in pigs, it was found that the bacterium induced a response of CD_4_ (+) lymphocytes followed by an infiltration of CD_8_ (+) lymphocytes, in addition to mast cells, related to the pathophysiological events. Despite the clear immunological infiltration, the infection continued its course. This cellular infiltrate triggered pathology in the lung, with the appearance of lesions and an increase in the number of bacteria in the tissue, which suggests that the developed response did not stop the infection [38].

Enzootic pneumonia is clearly a delayed hypersensitivity (delayed-type hypersensitivity: DTH) that occurs in the lung, and from the experimental point of view, it occurs in two distinct stages: in the first well-known sensitization, where T lymphocytes are activated, expanded, and differentiated into memory TCD_4_ (+) and Th1 CD_4_ (+), secreting IL-2, IL-12, and IFNγ recruit macrophages in the area of infection; in the second stage of provocation (elicitation), the process begins with the re-exposure of the antigen recognizing the memory Th1 CD_4_ (+); the result is the DTH response [39,40].

T_Reg_ lymphocytes are essential for maintaining peripheral tolerance, preventing autoimmune diseases, and limiting chronic inflammatory diseases. However, they also limit beneficial responses by suppressing sterilizing immunity and limiting antitumor immunity, and the hypothesis is that effector T cells may not be “innocent” parts in the suppressive process and, in fact, could enhance the function of T_Reg_ cells. The authors of [21,22,41] showed in their experiments with *Mycobacterium tuberculosis* that the protective immunity conferred by ThCD4 cells, in an adoptive transfer system, was able to control the disease with high efficiency only in the absence of T_Reg_ cells. In addition, the suppression of protection by the cotransferred T_Reg_ was not accompanied by a general increase in the expression of IL-10 or by a greater number of T CD4+ cells producing IL-10.

The production of proinflammatory cytokines increases inflammation in the lung, where the two stages of sensitization and provocation typical in DTH are being carried out, establishing a microcycle of activation, expansion, and cellular differentiation that recruits more mononuclear cells at the site of infection [39]. In the aforementioned study, we found that on Days 1 and 21 of blood sampling, the secretion of nine proinflammatory cytokines was evaluated: TNF-α, INF-α, INF-γ, IL-1β, IL-4, IL-6, IL-8, IL-10, and IL-12-p40. An increase in secretion was found in only five of them: TNF-α, INF-α, INF-γ, IL-10, and IL-12 p40 [35]. Although the inflammatory response is important in controlling pathogens, tissue injury and disease subsequent to *M. hyopneumoniae* infections seem to be caused more by the response of the host than by the organism itself [1,36].

*Mycoplasma hyopneumoniae* was recovered from the homogenates of the pneumonic lesions of pigs in Groups B, C, and D. In Group B, which was challenged and without treatment with *S. cerevisiae*, it was recovered in all pigs. However, when the tracheal explants were used, mycoplasma was recovered more frequently. The same occurred with the pigs in Groups C and D, where mycoplasma was recovered from four and six pigs, respectively [28].

The DWG was determined during the experiment in five stages: 1. previous; 2. start; 3. defined; 4. evolution; 5. final. The pigs in all the groups had similar DWG amounts in the first three stages: previous, start, and challenge. During stage 4, only the challenged groups showed a decrease in their weight, and during the evolution of the disease, only Group B showed a decrease in weight until the final stage of the experiment, while Groups C and D showed an increase in weight at this stage and a clear recovery from the disease.

These results suggest that live and disintegrated *S. cerevisiae* have an immunomodulatory effect in chronic proliferative pneumonia, DTH, in the pathogenesis of *M. hyopneumoniae*, where it seems to depend on the alteration or modulation of the respiratory immune response [1,42].

A field experiment carried out by Procajlo et al. [43] with animals vaccinated against *Mycoplasma hyopneumoniae* and challenged in a natural, uncontrolled manner showed that *Sacharomyces cerevisae* administered in food plus vaccination had an immunomodulatory effect in the prophylaxis of mycoplasmic pneumonia swine. While the animals in our experiment were not vaccinated or treated with antibiotics, the results of the clinical study, including the degree of pneumonic lesion and productive parameters, revealed that the group of pigs that received the disintegrated product of *Sacharomyces cerevisae* behaved better than the animals in the group that received the live yeast. When they were challenged with *Mycoplasma hyopneumoniae* under controlled conditions, they showed protection against enzootic pneumonia; therefore, perhaps it could behave as an immunodulator, which continues to give us direction to continue with the research to corroborate this effect.

## Figures and Tables

**Figure 1 pathogens-13-00322-f001:**
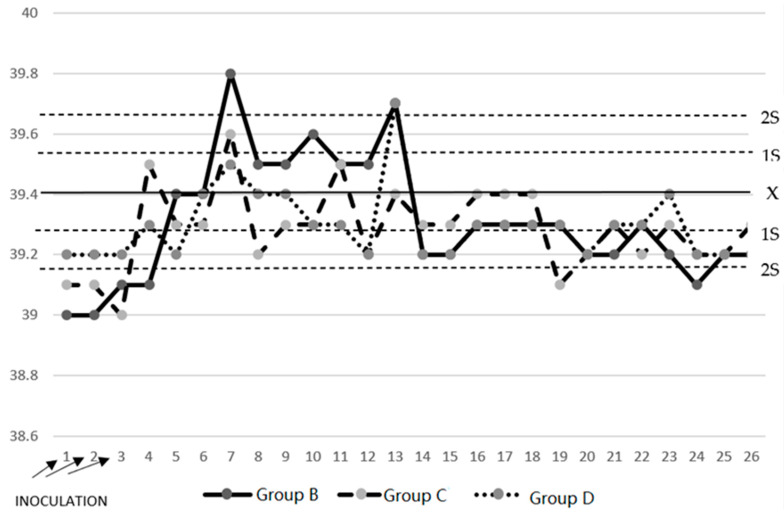
Average temperature of the experimental pig groups. Group B [ ]: aerosolized pigs during the first three days of the experiment with *Mycoplasma hyopneumoniae* strain 194. Note: bimodal hyperthermia in this group. Group C [ ] and Group D [ ] remained within normality. The continuous horizontal line represents the mean temperature (39.27 °C) of untreated or challenged Group A, and the dashed lines are located two standard deviations above (2S) and below (2S) the normal.

**Figure 2 pathogens-13-00322-f002:**
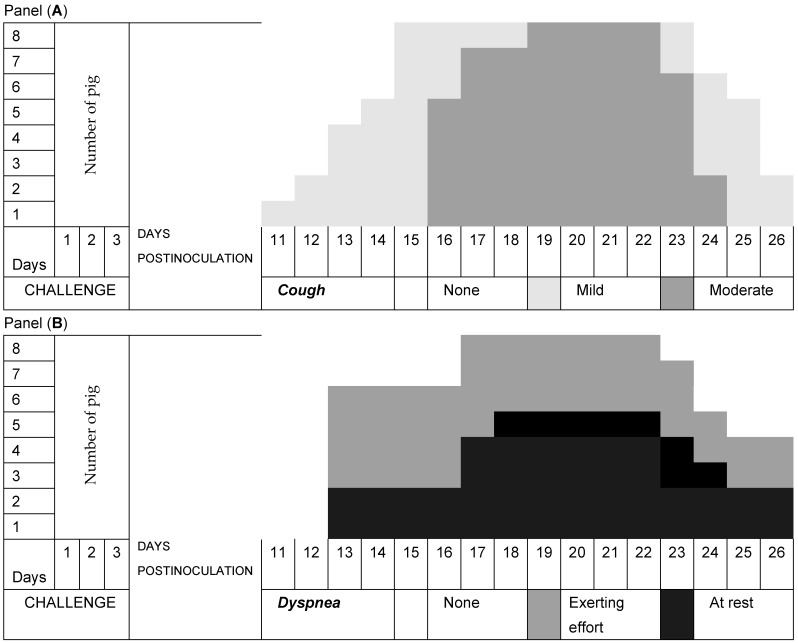
Evolution of respiratory signs. Group B was challenged with *Mycoplasma hyopneumoniae* only. Panel (**A**) shows the temporal duration and severity of the clinical signs of cough, while Panel (**B**) shows the temporal duration and severity of dyspnea.

**Figure 3 pathogens-13-00322-f003:**
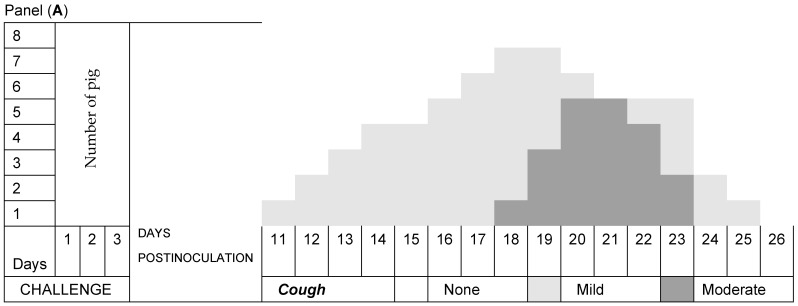
Evolution of respiratory signs. Group C was challenged with *Mycoplasma hyopneumoniae* and treated with the disintegrated yeast *Saccharomyces cerevisiae*. Panel (**A**) shows the temporal duration and severity of the clinical signs of cough, while Panel (**B**) shows the temporal duration and severity of dyspnea.

**Figure 4 pathogens-13-00322-f004:**
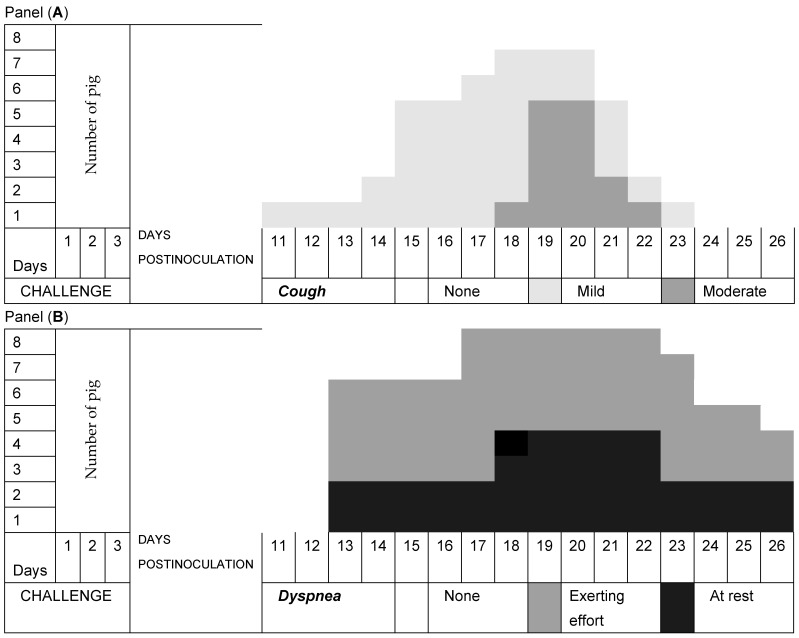
Evolution of respiratory signs. Group D was challenged with *Mycoplasma hyopneumoniae* and treated with the live yeast *Saccharomyces cerevisiae*. Panel (**A**) shows the temporal duration and severity of the clinical signs of cough, while Panel (**B**) shows the temporal duration and severity of dyspnea.

**Figure 5 pathogens-13-00322-f005:**
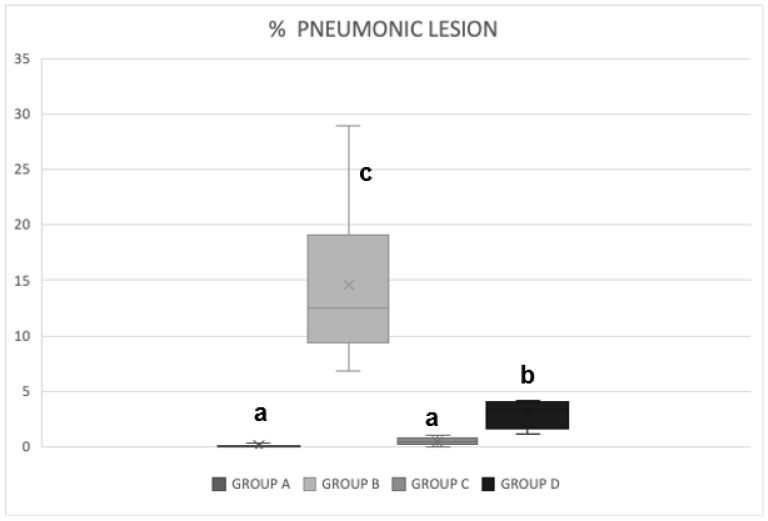
Extension of the pneumonic lesions found in the groups challenged with *Mycoplasma hyopneumoniae* and treated with disintegrated *Saccharomyces cerevisiae* and with live *Saccharomyces cerevisiae*. The pneumonic lesions found in Groups A and C were less extensive than those in Groups B and D (a: *p =* 0.05), while in Group D (b: *p =* 0.05), the lesions were less extensive than those in Group B (c: *p* = 0.01).

**Figure 6 pathogens-13-00322-f006:**
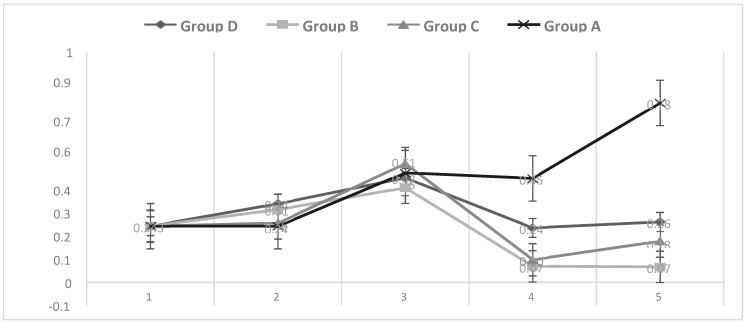
Daily weight gain (DWG), determined for the groups of experimental pigs: Group A, untreated and unchallenged. Group B, untreated and challenged pigs; Group C pigs treated with disintegrated and challenged *Sc*; and Group D, pigs treated with live and challenged *Sc*. The pigs in Groups B, C and D were aerosolized in the first three days of the experiment with *Mycoplasma hyopneumoniae* strain 194. The DWG was determined during the experiment in five stages: 1. previous; 2. start; 3. defined; 4. evolution and 5. final. *Sc*: *Saccharomyces cerevisiae*.

**Table 1 pathogens-13-00322-t001:** Recovery of *M. hyopneumoniae* from postmortem lungs.

Group ^a^ Treatment		Recovery of *M. hyopneumoniae*			
		Lung Homogenates		Bronchial Explants	
		Number of Positive Pigs	Mean Titer of Positives	Number of Positive Pigs	Mean Titer of Positives
**A**	None	0	0	0	0
**B**	*M. hyopneumoniae* alone	8	10^3^	8	10^5^
**C**	*Sc* disintegrated and *M. hyopneumoniae*	4	10^1^	4	10^3^
**D**	*Sc* live and *M. hyopneumoniae*	6	10^2^	6	10^3^

^a^ Each group contained eight pigs. Titer represents the number of color-changing units (CCUs) of *M. hyopneumoniae* in the lung tissue. Pigs aerosolized for 30 min with 10^4^ CCU/mL (10 mL/pig) on day and three times. *Sc*: *Saccharomyces cerevisiae*.

**Table 2 pathogens-13-00322-t002:** Location and appearance of macroscopic lung lesions ^a^.

Group	Consolidated Lobes							Appearance of Lesions		
	Apical		Cardiac		Diaphragmatic ^b^		Accessory	Network	Reddish Gray	Pleural Adhesions
	R ^c^	L ^c^	R ^c^	L ^c^	R ^c^	L ^c^				
**A**	**0**	**0**	**0**	0	0	0	0	0	0	0
B	8	8	8	8	7	6	3	2	6	2
C	3	3	2	2	2	2	1	2	1	0
D	6	6	6	6	3	3	1	4	1	0

^a^ Data indicate the number of animals affected in each group of eight. ^b^ Restricted to cephalic portions. ^c^ R = Right; L = Left.

## Data Availability

All data are provided in this article are in the Vega thesis.

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
