# Peer review of "Effect of Live and Fragmented Saccharomyces cerevisiae in the Feed of Pigs Challenged with Mycoplasma hyopneumoniae"

_pathogens, 2024, doi:10.3390/pathogens13040322_

Round 1
Reviewer 1 Report
Comments and Suggestions for Authors
This manuscript investigated the impact of probiotics in piglets' diets to mitigate the effects of Mycoplasma hyopneumoniae infection. It provides valuable insights into the immunomodulatory effects of Saccharomyces cerevisiae, contributing to the growing body of knowledge on alternative antimicrobial strategies in veterinary medicine.
The discussion could benefit from a broader comparison with related studies, especially in the context of the efficacy of different probiotic strains in similar settings. This would help to situate the study within the broader research landscape.
While the results are promising, the manuscript could further discuss the limitations of the study, including the scale of the experiment and the need for further research to confirm the findings and explore long-term effects and practical applications in pig farming.
The potential impact of the treatments on the microbiota of the piglets, beyond the scope of the immune response to Mycoplasma hyopneumoniae, is not extensively explored. Including this information could enhance understanding of the mechanisms behind the observed effects.
Line 17-19: “In a previous study, we found that on Days 1 and 21 of blood sampling, nine proinflammatory cytokines were secreted, and an increase in their secretion occurred for only five of them: TNF-α, INF-α, INF-γ, IL-10 and IL-12 p40.”, If it is not the result of this study, how can it be written in the abstract?
Figure 1: the line of Group D disappeared between d 13 and d 20. It is impossible to know which line is blocking this line.
Figure 5: What are the differences between a, b and c marked in the figure? And these letters are a little messy.
Table 1 and 2: Could the authors use chi-square test or other statistical method to statistically analyze the differences between groups?
Figure 6: Without legend, which group does each line represent?
Author Response
Thank you so much for your suggestions and corrections, we try to do them all.

Reviewer 2 Report
Comments and Suggestions for Authors
In the manuscript "Effect of live and fragmented Saccharomyces cerevisiae in the feed of pigs challenged with Mycoplasma hyopneumoniae", the authors describes the immunomodulatory effect of this specific agent in chronic proliferative M. hyopneumoniae pneumonia.
Please consider these suggestions for improving this manuscript:
Minor
Materials and Methods
Line 83: Did you check the absence of Abs against M. hyopneumoniae before moving the piglets to the isolation units?
Line 90: Is there any specific reason you administered different quantities of the yeast depending on its condition (Sc disintegrated 0.25 g/animal/day - Sc live 0.50 g/animal/day). I suggest you should provide some references or explain that difference.
Discussion
Lines 260 - 263: This part appears exactly the same in Introduction. You should use it either on discussion or introduction section.
In the discussion section, there is no mention of the potential variations in outcomes when employing disintegrated Sc versus live Sc. It's imperative to address these differences in results based on the Sc condition utilized. For example, Sc disintegrated or Sc live would be the choice of the Authors? Under what circumstances?
Author Response
Thank you for your suggestions and corrections, we try to do them all.
